# Determination of an Acceptable Portion Size of *Daal* for a Bangladeshi Community-Based Iron Intervention in Adolescent Girls: A Feasibility Study

**DOI:** 10.3390/nu13041080

**Published:** 2021-03-26

**Authors:** Fakir Md. Yunus, Chowdhury Jalal, Gordon A. Zello, Kaosar Afsana, Albert Vandenberg, Diane M. DellaValle

**Affiliations:** 1College of Pharmacy and Nutrition, University of Saskatchewan, 104 Clinic Place, Saskatoon, SK S7N 2Z4, Canada; fakir.yunus@usask.ca (F.M.Y.); gordon.zello@usask.ca (G.A.Z.); 2Nutrition International, 180 Elgin Street, Suite 1000, Ottawa, ON K2P 2K3, Canada; cjalal@nutritionintl.org; 3James P Grant School of Public Health, BRAC University, 68 Shahid Tajuddin Ahmed Sharani, Mohakhali, Dhaka 1212, Bangladesh; kaosar.afsana@bracu.ac.bd; 4College of Agriculture and Bio-Resources, The University of Saskatchewan, Agriculture Building 51 Campus Drive, Saskatoon, SK S7N 5A8, Canada; bert.vandenberg@usask.ca; 5Department of Sports Medicine, King’s College, 133 N River St, Wilkes-Barre, PA 18711, USA

**Keywords:** lentils, fortification, iron, adolescent girls, portion size, feeding study, Bangladesh, crossover trial, feasibility

## Abstract

Widely consumed *daal* (lentils) in Bangladesh are an ideal vehicle for iron (Fe) fortification; however, an acceptable portion size in meals needs to be determined to carry out a community feeding study in at-risk adolescent girls. A non-randomized crossover trial was conducted with *n* = 100 Bangladeshi girls (12.9 ± 2.0 years of age). Two recipes (thin and thick) and three portion sizes (25 g, 37.5 g, 50 g of raw lentil) of *daal* were served with 250 g of cooked white rice in a counter-balanced manner over 12 weeks. Each meal was fed to participants 5 days/week for two weeks. Ratings of hunger, satiety, and palatability were measured before and after each meal using Visual Analog Scales (VAS). The thick preparation in the 37.5 g portion (~200 g cooked) elicited higher VAS ratings of hunger, satiety, and palatability compared to all other meals. The 50 g portion of the thin preparation had VAS ratings similar to those of the 37.5 g thick preparation. Consuming the 37.5 g portion of fortified *daal* would provide 6.9 mg Fe/day to girls in a community-based effectiveness study. This would meet ~86% and ~46% of the Recommended Dietary Allowance (RDA) for Fe for girls aged 9–13 and 14–18 years, respectively.

## 1. Introduction

Iron deficiency anemia (IDA) is the most widespread nutritional deficiency in the world [1]. Annual deaths directly linked to IDA are around 24,000 globally and have been steadily increasing since 2000 [2]. The WHO 2011 report suggested that approximately 25% of the global population (~1.6 billion) were suffering from anemia, and preschool children and women of reproductive age contributed the major portion of this global burden [1]. In Bangladesh, a nationally representative survey (2013) reported that anemia prevalence (hemoglobin <12.0 g/dL) was 26% among non-pregnant non-lactating women and 17% adolescents aged 12–14 years [3]. Also IDA prevalence (hemoglobin <11.5 g/dL and ferritin < 5.0 μg/L) was 1.3% among children 6–11 years of age and 1.8% among adolescents aged 12–14 years [3]. However, iron deficiency (ID) prevalence without anemia (hemoglobin >12 g/dL and serum ferritin <15.0 μg/L) was reported to be slightly higher than anemia prevalence, i.e., 7.1% among non-pregnant non-lactating women and 9.5% among children aged 12–14 years. Several other studies have recommended Fe fortification as a preferable choice to reduce IDA [4,5,6]. In a systemic review, studies found that hemoglobin significantly increased among women of reproductive age due to the consumption of Fe-fortified foods [7]. 

We hypothesized that fortifying lentil (*Lens culinaris Medik*) with Fe could be a potential long-term sustainable solution to combat global ID and IDA among adolescent girls. We chose lentil because it is a widely consumed staple food in Bangladesh and it has a high nutrient density [8]. Furthermore, several studies have recommended lentil as a potential food vehicle for fortification because of its low content of anti-Fe absorption compounds (e.g., phytate) and high content of Fe, even pre-fortification [9,10,11]. A community-based human effectiveness trial, however, is warranted to estimate the effectiveness of Fe-fortified lentil on the improvement of body iron status. We expect that food-based supplemental Fe from Fe-fortified lentils will improve body Fe status. To carry out such a trial, several crucial aspects needed to be addressed, including the portion size of the lentil meal, the frequency and duration of feeding, participants’ acceptability of different portion sizes, and cooking preparation of the lentil meal to be consumed. Therefore, the objective of the current feasibility study was to determine an acceptable portion size of a lentil meal for adolescent girls’ daily consumption, which was critical prior to designing a large-scale community-based effectiveness trial. 

## 2. Materials and Methods

### 2.1. Study Setting and Population 

Adolescent girls (10–17 year) who were members of Bangladesh Rural Advancement Committee (BRAC) Adolescent Clubs were invited to participate in the current study. BRAC is a multi-national non-government organization (NGO) that was ranked number one among non-governmental organizations in the world for the fifth consecutive year till 2020 [12]. One of its developmental programs is designed for adolescents, providing a platform and opportunity to socialize both in rural and in urban settings. Adolescent boys and girls can be members regardless of school attendance, marital or pregnancy status, or economic standing. Around 25–40 adolescents participate in a club, and clubs run 2 days per week in the evenings. The current feasibility study purposively chose 4 adolescent clubs of 2 upazilas (sub-districts of Sreepur and Tongi Pouroshava municipalities) of the Gazipur district. These clubs were chosen because they were well functioning with good attendance of adolescent girls. While this study targeted adolescent girls, we served the *daal* meal to all boys and girls who were members of BRAC adolescent clubs and who were willing to consume the meals. No one declined to consume the meal, and there was no exclusion criterion for participation. As per the study objectives, however, data were collected only from adolescent girls. No adolescent girls declined to participate or were excluded from the study for any other reason.

### 2.2. Study Design and Trial 

A crossover trial was carried out with *n* = 100 adolescent girls aged 10–17 years. Those girls who were active members of a BRAC adolescent club and were aged between 10 and 17 years were included in the study. We served 2 different cooking preparations or recipes (thin and thick) of *daal* (a lentil meal) and 3 different portion sizes (equal to raw amounts of 25 g, 37.5 g, and 50 g of lentils) for 12 weeks (Figure 1). These portion sizes were decided based on the amount of Fe that could potentially be delivered within those portion sizes over the longer community trial. 

Due to the nature of the crossover study design, each adolescent girl served as her own control. Referring to previous food intake studies, a sample size of *n* = 100 adolescent girls was our target [13]. A total of *n* = 100 adolescent girls were enrolled from the four adolescent clubs (*n* = 25 girls participated from each club). Standardized recipes were used for each of the six conditions, with the basic *daal* recipe being locally derived (Table 1). Cooks were recruited from local neighbourhoods adjacent to each of the respective adolescent clubs and trained for this study. They completed cooking in a local kitchen under the close supervision of trained research assistants. 

Each adolescent girl was served 1 portion of each *daal* preparation (thin 25 g, thin 37.5 g, thin 50 g, thick 25 g, thick 37.5 g, and thick 50 g) 5 days a week for 2 weeks. Each portion size of *daal* was served with 1 cup of cooked polished local white rice (~250 g), since *daal* is not usually eaten without cooked rice in Bangladesh. These meals were served as late afternoon snacks in the BRAC adolescent clubs, between 4 and 6 pm. At first, the thin preparation of 25 g was served, followed by thin 37.5 g, thin 50 g, thick 25 g, thick 37.5 g. and thick 50 g (Figure 2). Iron fortification was carried out at the Crop Development Centre (CDC) of the University of Saskatchewan, Canada. Small cotyledon lentils were chosen because of their naturally high content of iron compared to other available lentils (Podder et al., 2017). Lentils were fortified with 16 ppm NaFeEDTA Fe fortificant/100 g lentils, sprayed as a fine mist, which was absorbed through a polishing drum at a commercial lentil mill located in Saskatoon, Canada. Details of the iron fortification have been described previously [14,15]. 

### 2.3. Study Variables and Outcome Measurements 

Survey data were collected by trained research assistants. At first, adolescent girls and their respective parents/guardians were informed separately by the study research assistants on the purpose of the study, data collection process, risks and benefits of participating in the study and were asked to sign the informed written consent and assent form with the presence of a witness. Once both of the parties signed the forms (consent and assent), a copy of both documents were given to them. Detailed demographic information was collected via face-to-face interviews, and anthropometric measurements were taken using standardized methods and guidelines [16,17,18,19]. 

Subjective ratings of hunger, fullness, thirst, nausea, prospective consumption, and sensory characteristics were obtained before and after consumption of each *daal* meal, using paper-based Visual Analog Scales (VAS). VAS are a well-accepted tool that captures subjective attitudes or characteristics for which the probable response lies within a range of values that cannot be directly or easily captured [20]. This measurement tool is commonly used in human feeding studies for measuring hunger, thirst, prospective consumption, fullness, and nausea [21,22,23,24]. 

Portion sizes (grams) of each *daal* meal were measured before and after consumption of each *daal* meal using a ±1 g precision rechargeable digital scale (MEGA Super; Reg: (Dhaka) No 128005). The residual amount of *daal* was also recorded. Cooked rice was not weighed, but rather served in a standard weight (1 c = 250 g) and mixed with cooked *daal* (cooked rice is locally known as *baat*). The estimated consumed iron was calculated, referring to the iron content of 50 g of raw lentils indicated in a previous study [14]. The study reported 10 mg of iron present in 50 g of raw lentils, of which 3–3.5 mg from lentils and 6.5–7 mg from the iron fortificant. 

### 2.4. Statistical Analysis

Data were analyzed with SPSS Statistics for Mac, version 25 (SPSS Inc., Chicago, IL, USA) using a mixed linear model and repeated measures ANOVA (RMANOVA). We used RMANOVA for *daal* intake (in grams) and each of the VAS ratings (in mm) of hunger, thirst, prospective consumption, feeling full, and feeling nauseated to compare before and after consuming the six preparations (thin 25 g, thin 37.5 g, thin 50 g, thick 25 g, thick 37.5 g, and thick 50 g). We used a linear mixed model to examine the consumption of each of the six preparations of *daal*, with the fixed effect of served *daal* (cooked amount), in order to determine the effect of the served *daal* on girl’s consumption (in grams), with the random effect of upazila. The interaction of these 2 factors was tested for significance before examination of the main effects. A modified Bonferroni procedure was used for post-hoc pairwise comparisons of the means. Results were considered significant at *p* < 0.05. 

## 3. Results

The demographic characteristics of the sample are presented in Table 2. In total, data from *n* = 100 girls were analyzed, and we experienced no dropouts during the 12-week feeding trial, with only 5.2% of meals being missed by the adolescents. Mean (±SD) age and body mass index (BMI) were 12.9 ± 0.9 years and 17.1 ± 3.0 kg/m^2^, respectively, and the majority of the participants (64%) reported a regular menstrual cycle. 

### 3.1. Differences in Preference by Cooking Preparation

Hunger, thirst, and prospective consumption were found significantly when comparing the thin and the thick preparations of *daal*, both before and after the meal (Figure 3). However, the mean score of the prospective consumption was significantly lower for the thick *daal* preparations compared to the thin meal preparations. 

### 3.2. Difference in Preference by Portion Size

Significant differences were found in regard to the different portion sizes before and after *daal* consumption (Figure 4). We observed lower prospective consumption ratings corresponding to the bigger portion sizes, and higher ratings of fullness corresponding to the larger portion sizes. 

### 3.3. Difference in Preferences by Combined Portion Size and Cooking Preparation

Significant differences were observed for all six preparations of different size: condition 1 (thin 25 g raw lentils), condition 2 (thin 37.5 g raw lentils), condition 3 (thin 50 g raw lentils), condition 4 (thick 25 g raw lentils), condition 5 (thick 37.5 g raw lentils), and condition 6 (thick 50 g raw lentils) before and after the meals (Figure 5). Before meal, the thick 37.5 g raw lentils recorded the highest prospective consumption rating, and after meal, the thick 37.5 g raw lentils had the second-lowest prospective consumption rating. 

### 3.4. Residual Amount and Estimated Iron Intake

Differences in portion size, consumed amount, residual amount, and estimated iron content of the consumed amount for all six conditions are presented in Table 3. We found that high values for portion size ended with the higher consumption, and the higher residual amount regardless of the cooking preparation. Significantly lower residual amounts (4.8 ± 12.4 g, 17.7 ± 29.6 g, and 27.1 ± 45.9 g for 25 g, 37.5 g, and 50 g of thick *daal*, respectively) were observed for all thick *daal* preparations compared to f thin *daal* (29.1 ± 44.0, 36.5 ± 55.3, and 61.9 ± 79.9 for 25 g, 37.5 g, and 50 g of thin *daal*, respectively). This indicates higher consumption of thick preparations compared to thin preparations. The mixed model suggested that the portions of served *daal* (cooked) corresponding to thin 37.5 g, thin 50 g, thick 25 g, and thick 50 g significantly affected girl’s daal consumption; however, the served amount of thin *daal* of 25 g and of thick *daal* of 37.5 g did not significantly influence their consumption.

## 4. Discussion

We conducted a feasibility study to identify an acceptable portion size of Fe-fortified lentils and the preferred cooking preparation to be used in a community-based effectiveness trial [15]. We observed lower residual amounts of the thick preparations of *daal* compared to the thin preparations. The teen girls preferred the thick *daal* to the thin one and said they could eat more of it; consequently, they consumed more lentil from this less-soupy, thick preparation in a lower volume. 

Determining the feasible *daal* portion size that could be consumed in a month-long trial is not straightforward. Age, current lentil consumption, iron recommended dietary allowance (RDA), and Fe bioavailability of the used lentils are major factors in determining the portion size to serve for effect size calculations and implementation for Fe interventions. For instance, age plays an important role in determining the necessary Fe intake because of the different daily Fe requirements by age groups. The RDA of Fe for girls aged 9–13 years is 8 mg/day, plus additional 2.5 mg/day for those who have started to menstruate and 15 mg/day for girls aged 14–18 years [25]. Around 50% of our study population was aged 10–12 years, which means that for half of the population, the RDA was 8.0 mg/day plus an additional 2.5 mg/day for the girls who had started to menstruate. We categorized our study population into two age groups: 10–13 years and 14–18 years, based on the age-specific RDAs for iron.

Considering the estimated daily iron intake, 4.5–4.9 mg iron from a thick/thin 25 g portion-size of *daal* provided an estimated 61.2% of the RDA for Fe for girls aged between 10 and 13 years (based on the 8 mg RDA for Fe) and 32.6% of the RDA for Fe for girls 14–18 years old (based on the 15 mg RDA for Fe). This additional Fe intake from the iron-fortified lentils may not be sufficient for girls 14–18 years of age, since it only covers one-third of their daily iron RDA. The 37.5 g (thick/thin) portion size provided 6.7–6.9 mg of Fe, which covered approximately 86% (and 65.7% for those who had started to menstruate) of the RDA for Fe for adolescent girls aged 9–13 years and 46% of the RDA for Fe for girls aged 14–18 years. An estimated 8.6–9 mg Fe was ingested from a 50 g (thick/thin) meal, which exceeded the RDA for Fe set for younger girls aged 10–13 years (8 mg/day). 

The 50 g portion of the thin preparation had VAS ratings similar to those of the 37.5 g thick preparation. However, considering the age-specific RDA for Fe and consumed Fe, the 37.5 g portion size was considered reasonable for an effectiveness trial. An additional argument in favor of the 50 g portion size is that it provides an amount of iron equal to or slightly higher than the RDA for Fe, and Fe has high Tolerable Upper Intake Levels (ULs) for girls aged 9–13 years (40 mg) and 14–18 years (45 mg) [25]. 

Before making a conclusion on a reasonable portion size, it is necessary to evaluate the relative Fe bioavailability of the portion of lentil considered suitable. While lentils are a plant source of non-heme iron, fortifying lentil with NaFeEDTA increases Fe absorption by 79% and significantly reduces the absorption of Fe inhibitor compounds, such as phytic acid [26]. Additionally, we chose dehulled lentils for Fe fortification, because an earlier study reported that lentil seed coat removal increases the relative Fe bioavailability [27]. This is because ferric iron has the highest binding stability with EDTA (ethylene diamine tetra-acetic acid) and EDTA thus protects from binding to iron absorption inhibitors [28,29].

Since we experienced no dropouts over 12 weeks, this may mean that teen girls in this area are likely to continue to consuming fortified *daal* in similar settings; however, the small number of participants (*n* = 100) was easy to retain over 12 weeks within the four BRAC clubs, possibly because of the close monitoring of the study participants by the study research assistants and their regular connection with the adolescent girls. Furthermore, BRAC, which organizes the adolescent clubs, is an NGO accepted by the community. Uncertainty exists with regard to how adolescents might behave in other contexts and to studies with a larger number of participants (e.g., *n* = 1200). In regard to the diet of rural Bangladeshi adolescent girls, seven-day food frequency data reported that fish, meat, eggs, and lentils were consumed, on average, 3.4, 0.6, 1.1, and 1.7 days out of 7 days, respectively [30]. We could find detail dietary intakes of lentil by adolescent girls in Bangladesh; however, an earlier study reported that the average consumption of pulses in Bangladesh is 12 g/day/person [31]. 

We provided a standard small cup (250 g) of cooked rice (locally known as *baat*) to all intervention conditions because, culturally, *daal* is usually consumed with *baat*; however, since each subject served as her own control, any fluctuation of any one participant’s *daal* consumption due to *baat* would have likely affected equally all conditions. 

A major strength of the study is the design, as all subjects were equally exposed to all interventions, and each participant served as her own control. We served *daal* meals 5 days per week and considered 2 days as a “wash-out” period, since there could have been the chance to observe a ‘carry-over effect’ if we fed the girls 7 days per week. Additionally, we used standardized recipes. One of the major limitations of the study was that the order of the intervention conditions (thin 25 g, thin 37.5 g, thin 50 g, thick 25 g, thick 37.5 g, and thick 50 g) was not randomly assigned or counter-balanced to the four adolescent clubs. Since we used the convenience sampling technique, generalizability cannot be applied. We did not collect information about other factors that may influence ID, which could be useful for future trials, but this feasibility study was not designed to examine those variables. Furthermore, the girls consumed 86–89% and 90–97% of the served thin and thick *daal*, respectively; this may mean that their prior dietary intake during the day may have influenced this high consumption. 

## 5. Conclusions

In conclusion, a portion size of 37.5 g of raw lentil in a thick cooking preparation using a local recipe was acceptable to adolescent girls aged 10–17 years; therefore, a community-based effectiveness trial could be carried out over a longer period of time to examine the effect of this amount of *daal* on iron status. 

## Figures and Tables

**Figure 1 nutrients-13-01080-f001:**
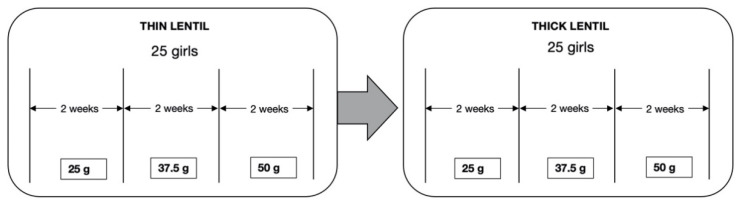
Cross-over design of the study that included 25 adolescent girls in each club (4 clubs × 25 adolescent girls = 100 adolescent girls).

**Figure 2 nutrients-13-01080-f002:**
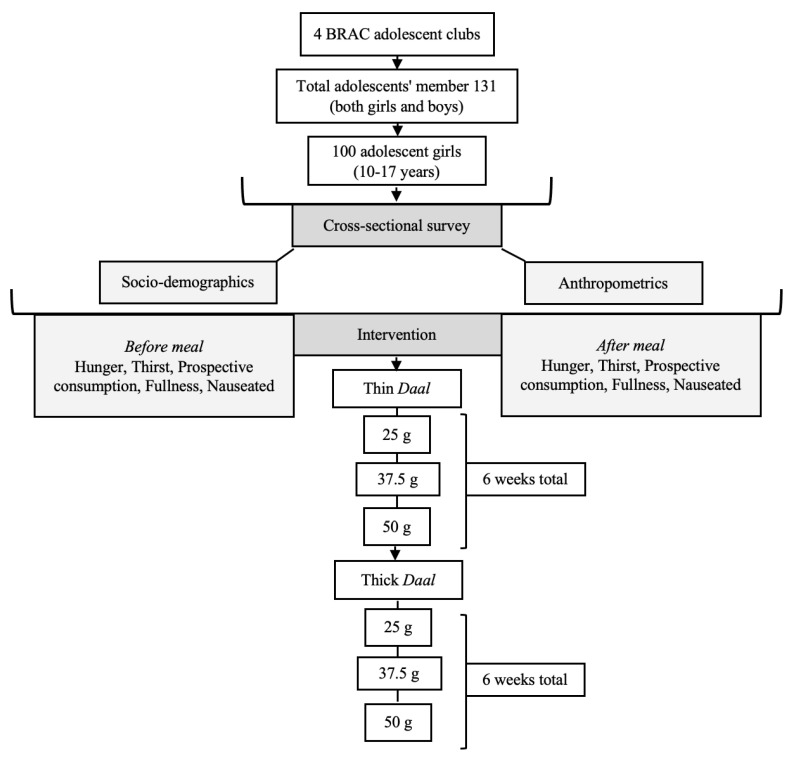
Study profile. BRAC, Bangladesh Rural Advancement Committee.

**Figure 3 nutrients-13-01080-f003:**
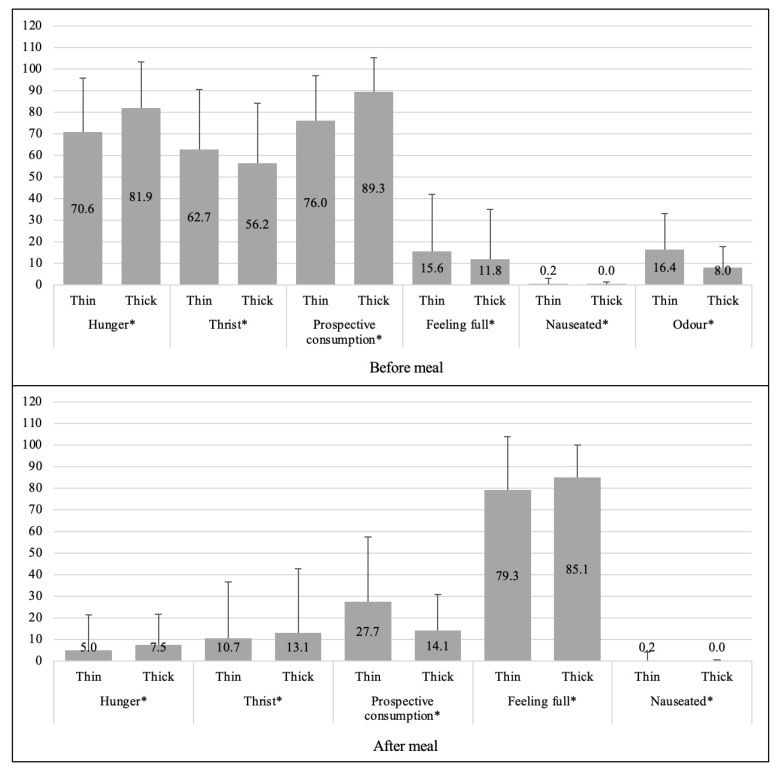
Mean ± SD differences of Visual Analog Scales (VAS) variables between thick and thin *daal* preparations before and after the meals (N = 100). Ratings of VAS scores: 0–100 mm, highest score = extremely happy, lowest score = not at all happy. The difference of VAS scores between thin and thick preparations of *daal* before and after the meals shows that in relation to thin and thick *daal*, preferences were significantly different for all VAS variables. This means adolescent girls had different preference in regard to thin and thick preparations of *daal* with respect to the parameters hunger, thirst, prospective consumption, feeling full, and feeling nauseated. * Independent sample t test significant at *p* < 0.05.

**Figure 4 nutrients-13-01080-f004:**
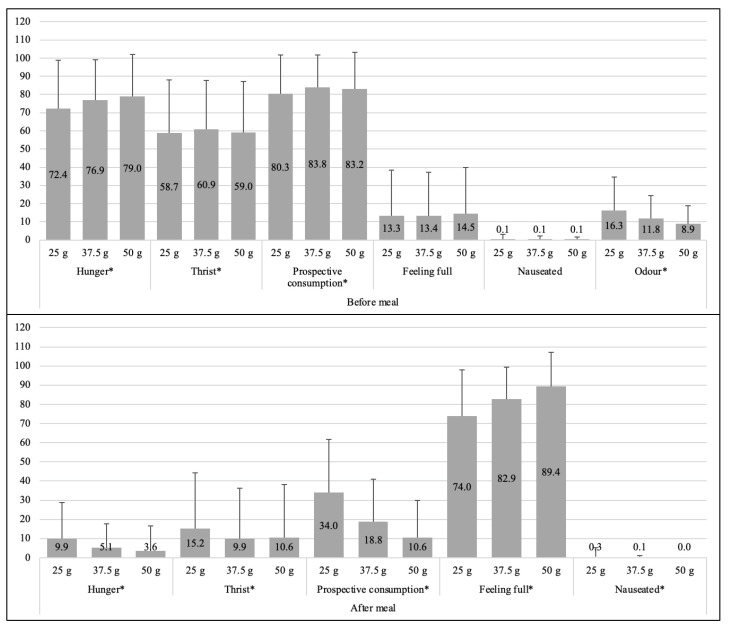
Mean ± SD differences of VAS variables among portion sizes before and after the meals (N = 100). Ratings of VAS scores: 0–100 mm, highest score = intense, lowest score = not at all intense. Differences of VAS scores for 25 g, 37.5 g, and 50 g of *daal* before and after the meals shows that preference for different portion sizes significantly differed with respect to each VAS variable. This means adolescent girls had different preferences for 25 g, 37.5 g, and 50 g of *daal* with respect to the parameters of hunger, thirst, prospective consumption, feeling full, and feeling nauseated. * One-way ANOVA test significant at *p* < 0.001.

**Figure 5 nutrients-13-01080-f005:**
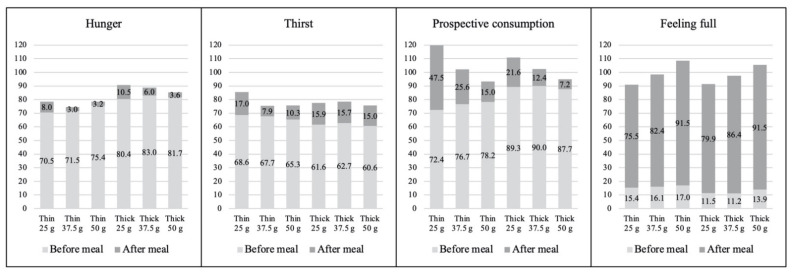
Difference in preferences in relation to the six portion sizes. Repeated measures ANOVA and post-hoc test with Bonferroni correction were used to assess significant differences for hunger after meal for the six preparations of cooked lentils: condition 1 (thin 25 g raw lentils), condition 2 (thin 25 g raw lentils), condition 3 (thin 50 g raw lentils), condition 4 (thick 25 g raw lentils), condition 5 (Before meal thick 37.5 g raw lentils), and condition 6 (thick 50 g raw lentils). Shown are significant differences of VAS scores for the six conditions of *daal* before and after the meals. Among all significant differences, we noted condition 5 (before meal, thick 37.5 g raw lentils) had the highest mean hunger score, followed by condition 6 (before meal, thick 50 g raw lentils). Condition 5 (before meal, thick 37.5 g raw lentils) had the highest prospective consumption score, followed by condition 4 (before meal, thick 25 g raw lentils). However, condition 12 (after meal, thick 50 g raw lentil) had the lowest prospective consumption score, followed by condition 11 (after meal, thick 37.5 g raw lentils). Condition 9 (after meal, thin 50 g raw lentils) and condition 12 (after meal. thick 50 g) had the highest score for sense of feeling full, followed by condition 11 (after meal, thick 37.5 g raw lentils).

**Table 1 nutrients-13-01080-t001:** Recipes of thick and thin preparations of iron-fortified lentils.

Sl. No	Thick Lentil Cooking Preparation and Amount (Based on 100 g of Uncooked *Daal*)		Thin Lentil Cooking Preparation and Amount (Based on 100 g of Uncooked *Daal*)	
	Ingredients	Amount (g)	Ingredients	Amount (g)
1	*Daal*	100	*Daal*	100
2	Turmeric	5	Turmeric	3
3	Chopped onion	40	Chopped onion	40
4	Garlic	8	Garlic	8
5	Green chili	3 smalls	Green chili	3 smalls
6	Water	700 mL	Water	1.5 L
7	Salt	1.5 teaspoonful	Salt	1.5 teaspoonful
8	Soyabean oil	10 teaspoonfuls	Soyabin oil	6 teaspoonfuls
9	Bay Leaf (Tejpata)	1 (small)	Bay Leaf (Tejpata)	1 (small)
10	Time	18 min	Time	53 min
	Weight after cooking: 585 g		Weight after cooking: 1166 g (1.166 Kg)	

**Table 2 nutrients-13-01080-t002:** Characteristics of the participants (*n* = 100)**.**

Variables	Adolescent Girls (Age 10–17 Years) %
Age (mean ± SD)	12.9 ± 2.0
Age of menarche (mean ± SD)	12.1 ± 0.9
Menstruation (Yes)	64
Menstrual regularity (once every month)	54
Marital status (No)	97
Education (Secondary)	67
Socioeconomic status	
House wall (Brick/Cement)	52
House roof (Tin)	95
House floor (Brick/Cement)	82
Electricity	98
Anthropometric	
Height; cm (mean ± SD)	146.8 ± 10.0
Weight; kg (mean ± SD)	37.3 ± 9.2
Waist circumference; cm (mean ± SD)	64.9 ± 7.0
Hip circumference; cm (mean ± SD)	78.1 ± 7.9
Mid-upper arm circumference (MUAC); cm (mean ± SD)	20.8 ± 3.2
Body mass index (BMI)	17.1 ± 3.0

**Table 3 nutrients-13-01080-t003:** Mean ± SEM differences for all conditions regarding served, residual, consumed *daal* meals and estimated iron consumption.

Consumption *Daal* Conditions	Served *Daal* (g)	Residual *Daal* (g)	Consumed *Daal* (Cooked Amount) (g)	Iron Contents in Consumed *Daal* ** (Mg)
Thin 25 g	274.2 ± 2.2 *	29.0 ± 4.4 * ^Ψ^	245.0 ± 4.9 *	4.5 ± 0.01 *
Thin 37.5 g	331.6 ± 1.5 *	36.2 ± 5.5 * ^Ψ^	294.8 ± 6.3 *	6.7 ± 0.1 * ^Ψ^
Thin 50 g	431.2 ± 1.6 *	61.5 ± 8.1 *	369.2 ± 8.2 *	8.6 ± 0.1 * ^Φ^
Thick 25 g	157.9 ± 0.7 *	4.9 ± 1.2 *	153.1 ± 1.6 *	4.9 ± 0.0 *
Thick 37.5 g	202.7 ± 1.5 *	17.5 ± 3.0 *	185.2 ± 2.4 *	6.9 ± 0.1 * ^Ψ^
Thick 50 g	256.4 ± 1.9 *	27.1 ± 4.6 * ^Ψ^	229.3 ± 4.5 *	9.0 ± 0.2 * ^Φ^

* Pairwise comparison with Bonferroni adjusted post-hoc analysis, significant at *p* < 0.001; ^Ψ^ and ^Φ^ Not significant; ** 50 g of fortified raw lentil = 10 mg of iron after cooking [14].

## Data Availability

Corresponding author may be contacted by qualified researchers for data sharing. All aspects of ethical issues and existing data sharing policies will be reviewed before sharing data.

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
