# Peer review of "Determination of an Acceptable Portion Size of Daal for a Bangladeshi Community-Based Iron Intervention in Adolescent Girls: A Feasibility Study"

_nutrients, 2021, doi:10.3390/nu13041080_

Round 1
Reviewer 1 Report
Dear Authors:
Although I found some merits in the study, additional parameters related to hunger/satiety (such as ghrelin, leptin) and also parameters related with iron metabolism (serum iron, TIBC, ferritin, transferrin, hemoglobin, hematocrit…) should have been assessed in order to strengthen the conclusions made by the authors, together with a comprehensive detail of the girl´s diet, to link these with the conclusions made. The manuscript can be interesting to a wide range of professional readers, effectively conveys the main findings, it is understandable, well structured and all sections are interesting and relevant. Ethical issues are correctly identified in the manuscript. It provides sufficient information to support the arguments being made and the methods used are correct. On the other hand, the conclusions of the paper are speculative, because the authors have not assessed iron status or iron metabolism. Another concern about the validity of the study is the maternal diet. This is a major flaw of the study, a source of bias and also a confounding factor in this type of studies. The authors have to supply data from the girl´s diet because obviously iron metabolism and status are highly linked to the diet and discuss this in the discussion section. Without knowing the characteristics of the girl´s diet, any conclusion made by this study is not valid at all.
Author Response
Reviewer 1:
Comments and Suggestions for Authors
Dear Authors:
Although I found some merits in the study, additional parameters related to hunger/satiety (such as ghrelin, leptin) and also parameters related with iron metabolism (serum iron, TIBC, ferritin, transferrin, hemoglobin, hematocrit…) should have been assessed in order to strengthen the conclusions made by the authors, together with a comprehensive detail of the girl´s diet, to link these with the conclusions made.
- Thank you for the comments. We agree that these parameters would strengthen the study conclusion; however, the study was aimed not to determine the portion size based on adolescent girl’s relative iron bioavailability using iron biomarkers rather this study finds out what would be the portion size that adolescents prefer to consume and would continue to consume under the community settings if we carry out a large-scale community-based efficacy trial. We determined the options of portion sizes (25 g, 37.5 g and 50 g raw amount) based on its iron contents and relative bioavailability that we found in our earlier study findings. We cited those accordingly. Also to share that this study was a methodological study determining the preferred portion size that to be used in the future effectiveness trial. L63-65
The manuscript can be interesting to a wide range of professional readers, effectively conveys the main findings, it is understandable, well structured and all sections are interesting and relevant. Ethical issues are correctly identified in the manuscript. It provides sufficient information to support the arguments being made and the methods used are correct.
- Thank you.
On the other hand, the conclusions of the paper are speculative, because the authors have not assessed iron status or iron metabolism.
- The objective of the study was to determine what would be the portion size that adolescents prefer to consume and whether they would continue to consume under the community settings if we carry out a large-scale community-based efficacy trial (L64-65). In the conclusion, we state that
“In conclusion, a portion size of 37.5 g of raw lentil in a thick cooking preparation using a local recipe was acceptable among adolescent girls aged 10-17 years, and could be feasibly to carry out a community-based effectiveness trial over a longer period of time to examine the effect on iron status.”
We believe that our conclusion is inline with the study objective.
Another concern about the validity of the study is the maternal diet. This is a major flaw of the study, a source of bias and also a confounding factor in this type of studies. The authors have to supply data from the girl´s diet because obviously iron metabolism and status are highly linked to the diet and discuss this in the discussion section. Without knowing the characteristics of the girl´s diet, any conclusion made by this study is not valid at all.
- We did not collect girl’s dietary data because it was not the study objective (L64-65). However, we added the followings in the discussion part L291
“Regards to the diet of the rural Bangladeshi adolescent girls, 7-day food frequency data reported that fish, meat, eggs and lentils were consumed on an average of 3.4, 0.6, 1.1 and 1.7 days respectively out of 7 days [31]. We could find detail dietary intake of lentil among the adolescent girls in Bangladesh; however, earlier study reported that average consumption of pulses in Bangladesh is 12 g/day/person [32].”
Reviewer 2 Report
This is an interesting piece of work in which the authors address the pertinent question of what is an acceptable amount of dal to provide iron in a population of adolescent and pre-adolescent girls - both relatively high risk groups for ID/IDA- that should be helpful in their future studies.
My biggest reservation is that the authors, in the body of the manuscript, provided no information on consent in this population, and ethics approval in the methods. I fully expected that they obtained consent and had ethics approval, and indeed found it as a post script - but it belongs in the methods.
I also have some questions about the statistics which will follow shortly, and here follow some questions about specific points.
Lines 46-48 - can the authors please clarify when they are discussing ID, and when they are discussing IDA, and to which the references relate
Study setting - please check grammar and English language through out this paragraph
What do the authors mean when they state that clubs were chosen to be well-functional (and check grammar), and then go on to say that club functionality did not make any difference
Was this a purposively chosen population - if so it should be made clear in the methods
Was the total population of girls 100? i.e. did the authors have 100% participation?
line 100 - check grammar
Can the authors explain how they used the mixed linear model with repeated measures?
In Figure 3, bottom graph, is the hunger difference score between thick and thin really statistically significantly different?
Figure 4- can the authors clarify what they mean by amongst all VAS variables?
Figure 5 legend mentions ANOVA - that should also be mentioned in the methods
Table 3 - the other data is SD, why is SEM used in Table3? Why is not all the data presented as SEM? Can the authors explain or make all SEM for consistency or if they feel SD is appropriate explain why
For table three in the legend what do the two different p's mean?
line 219-220 - please check grammar
line 237-238 - grammar
how does NaFeEDTA reduce physic acid?
lines 279-281 - how do the authors deduce that dietary intake influenced results?
Author Response
Reviewer 2:
Comments and Suggestions for Authors
This is an interesting piece of work in which the authors address the pertinent question of what is an acceptable amount of dal to provide iron in a population of adolescent and pre-adolescent girls - both relatively high risk groups for ID/IDA- that should be helpful in their future studies.
- Thank you.
My biggest reservation is that the authors, in the body of the manuscript, provided no information on consent in this population, and ethics approval in the methods. I fully expected that they obtained consent and had ethics approval, and indeed found it as a post script - but it belongs in the methods.
- We followed the journal format on stating the ethical aspect of the study. However, we agree that few lines on the consent and assent process would be useful. We added the following details of consent and assent process under the method section in L128
“At first, adolescent girls and their respective parents/guardians were informed separately by the study research assistants on the purpose of the study, data collection process, risk and benefit of participating of the study and asked to sign the informed written consent and assent form respectively with the presence of a witness. Once both of the parties signed the forms (consent and assent, respectively) a copy of both documents were given to them.”
I also have some questions about the statistics which will follow shortly, and here follow some questions about specific points. Lines 46-48 - can the authors please clarify when they are discussing ID, and when they are discussing IDA, and to which the references relate
- We revised the line since it was confusing L47 “Several other studies have recommended Fe fortification as a preferable choice to reduce IDA”
Study setting - please check grammar and English language through out this paragraph
- We revised the English grammar as advised.
What do the authors mean when they state that clubs were chosen to be well-functional (and check grammar), and then go on to say that club functionality did not make any difference
- Sorry about it. We deleted the text.
Was this a purposively chosen population - if so it should be made clear in the methods
- We stated sampling technique in the methods section L77 “The current feasibility study purposively chosen 4 adolescent clubs of 2 upazilas (sub-districts of Sreepur and Tongi Pouroshava municipality) of the Gazipur district. It was because these clubs were well-functional with good attendance of adolescent girls.”.
Was the total population of girls 100? i.e. did the authors have 100% participation?
- Yes, there was no drop out of participants; however, there were some meal missed during the 12 week study duration. We mentioned it in 173-74
line 100 - check grammar
- Corrected
Can the authors explain how they used the mixed linear model with repeated measures?
- We revised the Stat analysis subsection as follows L 153:
“Data were analyzed using SPSS Statistics for Mac, version 25 (SPSS Inc., Chicago, Ill., USA) using a mixed linear models and repeated measures ANOVA (RMANOVA). We used RMANOVA for daal intake (in grams) and each of the VAS ratings (in mm) of hunger, thirst, prospective consumption, feeling full and nauseated to compare before and after consuming the six conditions (thin 25 g, thin 37.5 g, thin 50 g, thick 25 g, thick 37.5 g and thick 50 g). We conducted the linear mixed models on the consumption of each of the six conditions of daal consumption with the fixed effect of served daal (cooked amount) to determine the effect of the served daal on girl’s consumption (in grams) holding the random effect of upazilla. The interaction of these 2 factors was tested for significance before examination of the main effects of the factors. A modified Bonferroni procedure was used for post hoc pairwise comparisons of means. Results were considered significant at p <0.05.”
We further added the followings in the 3.4 result sub-section L231 “Mixed model suggested that served daal (cooked) of thin 37.5 g, thin 50 g, thick 25 g and thick 50 g significantly affected girl’s daal consumption; however, served amount of thin 25 g and thick 37.5 g were not significantly influenced their consumption.
In Figure 3, bottom graph, is the hunger difference score between thick and thin really statistically significantly different?
- Yes, we re-run the syntax and it was sig. different as mentioned in the Fig 3.
Figure 4- can the authors clarify what they mean by amongst all VAS variables?
- Thank you for pointing this out. We revised the sentence (L201) as ‘significantly differs within each of the VAS variables.’
Figure 5 legend mentions ANOVA - that should also be mentioned in the methods
- We included the word L154 ‘ANOVA’ under method section in stat. analysis subsection.
Table 3 - the other data is SD, why is SEM used in Table3? Why is not all the data presented as SEM? Can the authors explain or make all SEM for consistency or if they feel SD is appropriate explain why
- We used SD in Table 2 because we understand that generally readers would be interested to know the variability within the sample. We used SEM in the Table 3 stats because it quantifies the uncertainty in estimate of the mean i.e how far the sample mean (average) of the data is likely to be from the true population mean.
For table three in the legend what do the two different p's mean?
- We are sorry about it. We re-run the syntax and checked again. We deleted p<0.05
line 219-220 - please check grammar
- Corrected
line 237-238 – grammar
- Corrected
how does NaFeEDTA reduce physic acid?
- We added the following text L275 “It is because ferric iron has the highest stability to bind with EDTA (Ethylene diamine tet-ra-acetic acid) and it acts as a protector from binding the iron with iron absorption inhibi-tors [27,28].”
lines 279-281 - how do the authors deduce that dietary intake influenced results?
- The objective of the study was to determine what would be the portion size that adolescents prefer to consume and whether they would continue to consume under the community settings if we carry out a large-scale community-based efficacy trial. The study was not designed to measure the effect of dietary intake on girls consumption.
Round 2
Reviewer 1 Report
The manuscript has not been improved and still has the same flaws as in the original version. The authors have not included any new determinations or assessments related to iron metabolism or the girls’ diet.
Author Response
The manuscript has not been improved and still has the same flaws as in the original version. The authors have not included any new determinations or assessments related to iron metabolism or the girls’ diet.
After careful discussion with the authors, we came to the consensus that the study was carried out appropriately and fulfilled the purpose. Again, the study was not aimed to conclude anything related to the iron biomarkers, Fe intake or content of girls' diets, or iron bioavailability. Rather, as stated, we set the study’s objective to determine the ‘acceptable portion size of daal’ that might be feasible to carry out a community-based effectiveness trial over a longer period of time to examine the effect of fortified lentils on iron status.
Reviewer 2 Report
I am happy with the updated version and look forward to the further study to evaluate if the dal decreases ID/IDA
The authors will want to carefully review the manuscript for some minor grammatical errors
things I spotted were in
lines 76-82
lines 154-156 - maybe just hard to read with all the strike throughs?
lines lines 267-269 - I think the added lines about iron EDTA are in the wrong place? do they belong in the next paragraph?
lines 223-224
line 312
Author Response
I am happy with the updated version and look forward to the further study to evaluate if the dal decreases ID/IDA
- Thank you.
The authors will want to carefully review the manuscript for some minor grammatical errors
- We thoroughly checked the manuscript again. And correctly accordingly.
Things I spotted were in
Lines 76-82 - Corrected
Lines 154-156 - Corrected
Lines lines 267-269 - I think the added lines about iron EDTA are in the wrong place? Do they belong in the next paragraph?
- Yes, we corrected it. Many thanks.
Lines 223-224 - corrected
Line 312 - corrected